Plantar pressure distribution of ostrich during locomotion on loose sand and solid ground

Zhang Rui zhangrui@jlu.edu.cn
Han Dianlei
Ma Songsong
Luo Gang
Ji Qiaoli
Xue Shuliang
Yang Mingming
Li Jianqiao
Key Laboratory of Bionic Engineering, Ministry of Education, Jilin University , Changchun , People’s Republic of China
Hutchinson John
Electronic publication date: 2017 Jul 25
Publication date: 2017
Volume: 5
Electronic Location ID: e3613
Received 2016 Aug 15; Accepted 2017 Jul 5
Copyright: ©2017 Zhang et al.
Copyright year: 2017
Copyright holder: Zhang et al.
License: This is an open access article distributed under the terms of the Creative Commons Attribution License, which permits unrestricted use, distribution, reproduction and adaptation in any medium and for any purpose provided that it is properly attributed. For attribution, the original author(s), title, publication source (PeerJ) and either DOI or URL of the article must be cited.
License URL: https://creativecommons.org/licenses/by/4.0/

Keywords: Ostrich, Plantar pressure distribution, Loose sand, Solid ground

Funding: National Natural Science Foundation of China 51675221 51275199 Science and Technology Development Planning Project of Jilin Province of China 20140101074JC This study was funded by the National Natural Science Foundation of China (No.51675221, 51275199), and the Science and Technology Development Planning Project of Jilin Province of China (No. 20140101074JC). The funders had no role in study design, data collection and analysis, decision to publish, or preparation of the manuscript.

==============================
Background

The ostrich is a cursorial bird with extraordinary speed and endurance, especially in the desert, and thus is an ideal large-scale animal model for mechanic study of locomotion on granular substrate.

Methods

The plantar pressure distributions of ostriches walking/running on loose sand/solid ground were recorded using a dynamic pressure plate.

Results

The center of pressure (COP) on loose sand mostly originated from the middle of the 3rd toe, which differed from the J-shaped COP trajectory on solid ground. At mid-stance, a high-pressure region was observed in the middle of the 3rd toe on loose sand, but three high-pressure regions were found on solid ground. The gait mode significantly affected the peak pressures of the 3rd and 4th toes (p = 1.5 × 10−6 and 2.39 × 10−8, respectively), but not that of the claw (p = 0.041). The effects of substrate were similar to those of the gait mode.

Discussion

Ground reaction force trials of each functional part showed the 3rd toe bore more body loads and the 4th toe undertook less loads. The pressure distributions suggest balance maintenance on loose sand was provided by the 3rd and 4th toes and the angle between their length axes. On loose sand, the middle of the 3rd toe was the first to touch the sand with a smaller attack angle to maximize the ground reaction force, but on solid ground, the lateral part was the first to touch the ground to minimize the transient loading. At push-off, the ostrich used solidification properties of granular sand under the compression of the 3rd toe to generate sufficient traction.

Introduction

The effective locomotion of terrestrial animals depends on the interaction between the functional body parts and the substrate (Dickinson et al., 2000). Evolution has pushed their environmental adaptation, so the limbs of animals living in different environments show various functional morphology and kinematics. Even the same animal may adopt substrate-specific locomotion strategies. For example, when running on solid surface, the desert-dwelling zebra-tailed lizard (Callisaurus draconoides) uses a digitigrade foot posture to recover about 40% of the whole mechanical work (Li, Hsieh & Goldman, 2012), but when running on granular surface, the lizard employs a plantigrade foot posture to paddle through granular substrate and generate enough force for acceleration (Li, Hsieh & Goldman, 2012; Li, Zhang & Goldman, 2013).

Despite the complicated interaction between limbs and the ground, recent studies demonstrate both walking and running gaits emerge from a spring-loaded inverted pendulum model, and the gait dynamics vary largely along with different combinations of kinetic energy, leg compliance and leg contact conditions, resulting in different oscillation modes (Dickinson et al., 2000; Geyer, Seyfarth & Blickhan, 2006; O’Connor, 2009). As for walking, it involves an inverted-pendulum model that accounts for the efficient exchange between potential energy and kinetic energy within each step (Cavagna & Kaneko, 1977; Alexander, 1991). Benefiting from this model, humans can not only save 70% of total mechanical work to move body mass at the optimal speed, but also carry up to 20% of body weight on the head without additional energy (Cavagna & Kaneko, 1977; Cavagna et al., 2002). Moreover, as for rapid locomotion such as running, hopping and galloping, the center of mass (COM) is inverse to the walking gait, as it minimizes at the mid-stance and maximizes at the touchdown and take-off. Therefore, these bouncing gaits are described as a spring-mass model that accounts for the exchange of mechanical energy. In the spring-mass model, mechanical energy is stored as elastic strain in spring-like tissues (e.g., tendons) at the first half of the stance phase, and then returned by elastic recoil at the second half (Cavagna & Kaneko, 1977; Alexander, 1991; Spence et al., 2010). Moreover, the spring-mass model could approximate the gait stability of a mammal or bird running on diverse terrains (Geyer, Seyfarth & Blickhan, 2005; Birn-Jeffery et al., 2014). Although the inverted-pendulum model and the spring-mass model both represent rather simple mechanical systems, these fundamental mechanisms have been observed in humans, birds, cockroaches and lizards, and are utilized to carry out the projects of various bionic robots (Birn-Jeffery et al., 2014; Geyer, Seyfarth & Blickhan, 2006; Full & Tu, 1990; Heglund et al., 1982; Collins et al., 2005).

These models all simplify the substrate as a rigid, flat and perfectly elastic ground, but in reality, natural substrates such as sand, soil and snow display both solid-like and fluid-like features (Li, Zhang & Goldman, 2013). During movement on natural substrates, solid-fluid transition happens beyond the yield stress and then results in increasing the energy cost and instability of gait (Lejeune, Willems & Heglund, 1989). Furthermore, comprehensive models are less well-developed to predict the reaction and leg kinematics on granular substrates than solid substrates. As we all know, soft substrates significantly affect the results.

In order to study locomotion on flowing substrates, researchers proved granular dry sand was an ideal substrate, because it displayed both solid-like and fluid-like behaviors and was simpler than other flowing substrates such as snow and soil (Maladen et al., 2009; Marvi et al., 2014). Many small and light desert animals, such as lizards, were utilized to study the locomotion mechanics on granular sand because of their high locomotion performance on a wide range of desert terrains (Korff & McHenry, 2011; Crofts & Summers, 2011; Irschick & Jayne, 1999). It was found natural variation in substrate mechanics did not greatly affect the sprinting performance of either C. draconoides or Uma scorparia.

Moreover, recent studies on the interaction between small legged robots and granular materials (e.g., sand) reveal the effective locomotion on granular materials depends on the solidification properties of the granular substrate (Li et al., 2009; Li et al., 2011). This dependence is partially attributed to the fact that below the yield stress, the forced granular substrate remains solid, but beyond the yield stress, it flows like a fluid. Based on the solidification mechanism, we can explain and predict the rapid legged locomotion on sand by using empirical models (Li, Zhang & Goldman, 2013). For example, sea turtles, only by using solidification properties of sand under the flippers, can crawl on loose sand at a speed three times the body length per second without any slip (Mazouchova et al., 2010; Mazouchova, Umbanhowar & Goldman, 2013).

Falkingham reconstructed the 3D foot movements of guineafowl traversing a granular substrate from biplanar X-rays, and then incorporated the kinematics into a discrete element simulation (Falkingham & Gatesy, 2014). As the soil thickness increased, footprints gradually became shallower and the force acting on sand also decreased. However, these changes can only be qualitatively described.

Relevant literatures are focused on small animals, which are convenient for trials. However, plantar pressures in large animals have not been reported. African ostrich (Struthio camelus) is an ideal model for locomotion research of large and heavy animals on natural surfaces, as this highly-specialized cursorial bird has extraordinary speed and endurance (Alexander et al., 1979; Abourachid & Renous, 2000). This large terrestrial bird moves faster than other ratites and has been filmed running at the speed of 60 km h−1 for 30 min, with a peak of 80 km h−1 (Alexander et al., 1979; Abourachid & Renous, 2000). Ostriches live in deserts, especially semiarid deserts with short grasses, and inhabit about 25 km2 territories (Williams et al., 1993). In order to survive in deserts, ostriches locomote efficiently on diverse substrates (e.g., sand, rocks and grass) to find enough food and water. Compared with bipedal humans, ostriches run at two times speeds with about 50% lower of metabolic cost, although both species have similar weight and height (Watson et al., 2010; Rubenson et al., 2011).

This efficient locomotion of ostriches is mechanically supported by the extended jointed chain system of their hindlimbs with the center of mass close to the hip, which maximizes the gravitational and elastic potential energy (Abourachid & Renous, 2000). Long and slender tendons, coupled with the flexion and extension of joints, store and return a large proportion of mechanical energy, which would otherwise be compensated by muscle power (Alexander et al., 1979; Rubenson et al., 2011; Smith et al., 2006). The permanently elevated metatarsophalangeal joint (MTP) (Fig. 1) is a highly-specialized structure for transforming kinetic energy and gravitational potential energy into elastic strain energy of muscle-tendon units, which accounts for the majority of elastic energy stored in this joint (Alexander et al., 1979; Rubenson et al., 2011). Furthermore, the longest hindlimb among ratites, in combination with proximally- concentrated muscles, endows ostriches with high stride frequency and large step lengths (Smith et al., 2006; Fowler, 1991).

Figure 1 MRI image of the ostrich foot with the MTP and digit cushion.

The adaptation of rapid locomotion can also be observed in ostrich foot toes. Each ostrich foot has only two digits (the 3rd and 4th toes) covered with closely adherent vertical cornified papillae, whereas other ratites have three digits with simple callous plantar pads (Fowler, 1991). The 3rd toe has two approximately parallel digit cushions (Fig. 1) that are fixed by fibrous connective tissues and extend from the 1st to the 4th phalanxes, while the 4th toe has only one cushion under the phalanxes (El-Gendy, Derbalah & El-Magd, 2011). A recent study on dynamic pressure distribution of ostrich locomotion on solid ground reveals that the 3rd toe bears the most loads of the body mass, the 4th toe is responsible for stabilization and the claw functions as a position anchor (Schaller et al., 2011). Despite many suggestions about ostrich running rapidly on various terrains, no study has examined the mechanism how the two toes interact with granular materials.

In this study, we explored the plantar pressure distributions of ostrich toes, and assessed the functions of toes when ostriches ran/walked on two typical and well-defined substrates (loose sand and solid ground) that were similar to the habitat environment of ostriches.

Materials and Methods

Animals

Ten healthy juvenile ostriches with healthy feet were selected from a local breeder. Ethical approval was given by the Animal Experimental Ethical Inspection, Jilin University (reference No. 3130089). These ostriches were kept outdoor. Each ostrich was trained to walk and run on a fenced-in corridor for at least 30 min each time, twice per day, and over a month before data collection. After comprehensive comparison of representation and amenability, we selected two tractable female adult ostriches to finish all tests. Other female ostrich feet were brought from a slaughter house and then scanned by magnetic resonance imaging (MRI). The three-dimensional reconstruction and MRI images of the feet are shown in Figs. 1 and 2, respectively.

Figure 2 Three-dimensional reconstruction of ostrich left foot and the model ostrich for trials.

(A) ventral view; (B) front view; (C) isometric view; (D) lateral view; (E) the model ostrich for trials with marks on its legs.

Trackway treatments

An 80-m-long and 2-m-wide runway was constructed near the animal enclosure and fenced by 1.5-m-tall meshes (Fig. 3). All pebbles or sundries were removed. Within the runway, a trackway was narrowed to 1.4 m wide and 4 m long where the experimental equipment was installed. A 6-m-long and 4-m-wide sunshade net was set 3 m above the trackway to prevent any mistakes induced by sunlight spot in digital reconstruction. In solid ground trials, a custom-designed polycarbonate frame (2 × 1.5 × 0.015m3) was made to fix the pressure plate. The encased pressure plate was placed on one level marble slab (2 × 1.4 × 0.03m3) to avoid bending, and covered with a rubber sheet (4 × 1.4 × 0.0025m3) to avoid potential damage. In loose sand trials, the trackway was prepared with a 4-cm-thick layer of loose and dry sand to simulate the living condition in farm and natural desert. The sand thickness was selected with some considerations. If the sand thickness was too large, the effect on plantar pressure would be much more blurry, and the plantar pressure distributions were unclear or even more inaccurate. If the sand thickness was too thin, the action of soft sands on ostrich foot would be ineffective. Therefore, the sand thickness of 4 cm was selected to preliminarily study the plantar pressures of ostriches. The pressure plate and its surrounding components were placed underground, and then the sands were laid on the rubber sheet until the sand surface was leveled with the ground surface. The particle size distribution of sands is shown in Fig. 4. Before each trial on loose sand, the sands were prepared and the level conditions were confirmed to make sure the trackway was in the same state.

Figure 3 Schematic of the trackway.

(A) Top view; the shade was the data capturing area covered with the sunshade net. (B) Experimental setup for capture of plantar pressure.

In each trial, each ostrich ran (or walked) on the loose sand (or solid ground) for at least three valid times, with more details shown in Table 1. The ostrich (1) walked on the solid ground, (2) ran on the solid ground, (3) walked on the loose sand, or (4) ran on the loose sand. Running and walking were classified as the velocity of 2.5–4.1 and 0.7–1.7 m s−1, respectively. Ground running was not studied in this study.

Plantar pressure distribution and ground reaction forces

Plantar pressure distribution and ground reaction forces were measured by an RSscan International pressure plate (2, 096 × 472 mm2, 500 Hz sampling, 16,384 sensors with 0.5 × 0.7 cm2, USBII interface; Olen, Belgium). Data were acquired and analyzed on Footscan7 Gait 2nd generation (RSscan International, Olen, Belgium). Then we divided an ostrich foot into three functional parts: the 3rd toe (without claw), the 4th toe and the claw (Fig. 5) and investigated the effects of each part on locomotion. The ground reaction force of each plantar part was plotted and normalized against body weight. Total ground contact area, the ground contact area of each functional part, and the angle between the 3rd and 4th toes were determined on Footscan7 Gait based on Fig. 5. After the ostrich moved across the sand trackway, we photographed its footprints using an ordinary digital camera. After the surface of the pressure plate was covered with a 4-cm-thick layer of sand, the data of sample weights were inputted into the computer. Then the sample moved on the pressure plate to complete the calibration. The pressure plate was recalibrated after each trial.

Statistical analysis

We used analysis of variance (ANOVA) to analyze the differences between two samples, and the effects of gait and media on plantar pressure. Under the condition of walking / running on solid ground / loose sand, we chose indicators of total peak pressure, peak pressure of 3rd toe, peak pressure of 4th toe and peak pressure of claw. Significance level was set at P < 0.05. The valid numbers of trials collected and of trials analyzed are shown in Table 1.

Figure 4 Particle size distribution of sand used in trackway.

Table 1 Tests of two samples under different media and gait modes.

	Solid ground	Loose sand	
	Walking	Running	Walking	Running	
Number of trials collected	10	17	6	11	
Number of trials analyzed	10	10	6	6	

Figure 5 Typical Footscan images of plantar surface of the foot in running on loose sand (A) and solid ground (B).

Claw, distal 3rd toe, arch, proximal 3rd toe and 4th toe corresponded to different foot parts. α was the angle between 3rd toe and 4th toe.

Results

Center of plantar pressure

When the ostrich ran on the loose sand, the COP mostly originated from the middle of the 3rd toe (Fig. 6A). The 3rd toe was the first part to touch the loose sand (16 ms), followed immediately by the 4th toe (63 ms) (Fig. 7A), so the COP trajectory moved towards the 4th toe. However, the key function of the 3rd toe was to bear the body weight, and an elliptical COP trajectory appeared in some trials at about the first half of the stance phase. The origin of COP indicates the middle of 3rd toe was the first to touch the sand, followed immediately by the 4th toe. At the last half of the stance phase, COP moved from the middle of the 3rd toe directly to the claw, but the 4th toe gradually left the loose sand (158, 221 and 285 ms; Fig. 7A).

Figure 6 Typical COP path (shown as black dots) of the ostrich right foot.

(A) Running on loose sand, (B) running on solid ground, (C) walking on loose sand, (D) walking on solid ground. Red, yellow, green and blue indicate the plantar pressure decreased gradually.

Figure 7 Roll-off patterns of the right toe during loose sand (A) and solid ground (B) running trials.

The images were recorded at 16, 63, 158, 221, and 285 ms after touch-down of foot. Loads were indicated on a gradient from red to dark blue (the same below).

Different from the COP trajectory observed in the loose sand trials (Fig. 6A), a J-shaped COP trajectory was found in the solid-ground running trials (Fig. 6B). The areas of ground touching show the 3rd and 4th toes touched the ground simultaneously (16 and 63 ms; Fig. 7B). Therefore, when an ostrich ran on the solid ground, the COP originated from the lateral part of the 3rd toe. During solid-ground running trials, COP followed a similar curve path that it moved from the lateral part to the longitudinal center of the distal 3rd toe, and then straightly to the claw after about 70% of the stance phase (Fig. 6B).

In most running trials, the course of COP traversed a longer distance and was closer to the arch on loose sand than on solid ground, which was because the arch contacted with the sand and carried a part of the load (Fig. 6A).

When the ostrich walked on loose sand or solid ground, the COP path originated from the similar position as that in the running trails. On the loose sand, the COP path was more changeable than the one on the solid ground (Fig. 6C). On solid ground, the distal 3rd toe and the 4th toe touched down almost simultaneously at the beginning of the stance phase (43 ms, Fig. 8B), which provided a stable landing and resulted in a high variable COP distribution at the initial 5% of the stance phase (Fig. 6D).

Figure 8 Roll-patterns of right toe during loose sand (A) and solid ground (B) walking trials.

The images were recorded at 43, 174, 435, 609 and 782 ms after touch-down of foot.

Plantar pressure distribution and roll-off pattern

Among all contact elements in the sand running trials, the 3rd toe was the first to touch down, followed by the 4th toe and the claw (Fig. 7A). During ground running trials, the 3rd toe was the first to touch the ground firs, followed by the claw almost immediately and finally the 4th toe (Fig. 7B). The sand walking trials involved the same roll-pattern as the sand running trials (Fig. 8A). During ground walking trials, the distal 3rd toe touched down at the beginning of stance phase, followed by the proximal 3rd toe and then the 4th toe (Fig. 8B).

During sand running trials, a high-pressure region in the middle 3rd toe was observed around the initial contact region at the mid-stance, suggesting the distal 3rd toe bore the most load (Fig. 7A). At toe-off, the sands under the claw carried partial load, which resulted in connecting images between the claw and the distal 3rd toe. Except for a smaller pressure, the plantar pressure distribution of loose-sand walking (Fig. 8A) is the same as that in the running trials. The footprint examination in Fig. 9 supported the pressure distributions. The sand was solidified and flatted under the high-pressure region of the 3rd toe. Furthermore, two distinct sand bulges appeared on the sand surface under the compression of the arch and the claw.

Figure 9 Footprints of ostrich right foot when (A) running and (B) walking on loose sand.

The black solid lines show the region of disturbed sands. The dashed lines show the region of sand solidified and flatted by different parts of ostrich foot. An obvious hump appeared between the distal and proximal parts of the 3rd toe in both running and walking trials.

When the ostrich ran on solid ground, almost all parts of toes contacted the ground at the same time. As showed in Fig. 7B, three high-pressure areas on the 3rd toe and close to the digital joints were found after 20% of the stance phase. During walking trails, the dynamic pressure distribution was consistent with that in running trials except for the lower pressure and the exclusion of the claw from the initial contact period (Fig. 8B). Unlike on loose sand surface, the arch of ostrich foot could be easily identified from the plantar pressure profiles when the ostrich moved on the solid ground.

The surface areas on loose sand recorded by the pressure plate are shown in Table 2. Clearly, the total pressure area and the pressure areas of main supported parts of ostrich toes are slightly larger during running trails than walking trails. However, the areas of the claw were about 52.7% larger and the angle αwas 44.1% larger in running trials than in walking trials. When the trials were changed from ground walking to ground running, the load-bearing areas excluded the claw and α increased. The total surface area increased by 11.5% in running trails, which resulted from the 4.4% increase in the area of the 3rd toe and the 36% increase in the area of the 4th toe. In the loose sand trials, however, the area of the claw almost remained unchangeable when the bait was changed from walking to running. Total areas of toes on these two substrates suggest no obvious slip occurred, which was also supported by the qualitative analysis of videos and footprints.

Table 2 Plantar surface areas of total foot and functional parts.

	Total foot (cm2)	3rd toe (cm2)	4th toe (cm2)	Claw (cm2)	α (°)	
Running on loose sand	144.9 ± 7.6	89.9 ± 4.8	41.5 ± 4.2	11.3 ± 3.5	24.5 ± 6.7	
Walking on loose sand	133.5 ± 11.1	89.6 ± 5.5	34.9 ± 8.4	7.4 ± 2.7	17.0 ± 9.0	
Running on solid ground	159.8 ± 4.5	99.3 ± 3.9	48.4 ± 3.1	11.6 ± 1.7	20.3 ± 3.1	
Walking on solid ground	143.3 ± 6.5	95.1 ± 4.5	35.6 ± 5.1	11.1 ± 1.4	19.6 ± 4.9	
Notes.

Data are expressed as mean ± standard deviation (the same below).

Ground reaction force in running trials

The curves of total vertical ground reaction force were similar between different running trials, with a sharp increase at the first half of the stance phase and a quick decrease at the second half (Fig. 10A). The magnitudes of total ground reaction force on loose sand and solid ground were over 2.5 and 1.8 times of body weight, respectively. Among all contact elements during loose sand trials, the 3rd toe attained the maximum load at 50% of the stance phase (Fig. 10B), while the 4th toe reached the peak force at 35% of the stance phase and left the substrate at 69% of the stance phase (Fig. 10C). The forces of the claw were detected at 8% of the stance phase, increased to the peak at 80%, and lasted to the end (Fig. 10D).

Figure 10 Vertical ground reaction forces versus stance phase during loose sand and solid ground running trials.

Vertical ground reaction forces of (A) total foot, (B) 3rd toe, (C) 4th toe, and (D) the claw were all normalized against the body weight. The solid lines and dotted lines stand for the mean and standard deviations, respectively (the same below).

The loads of contact elements increased simultaneously during solid ground running trials. Firstly, the maximum load on the 4th toe was 34%, and then it shifted to the 3rd toe which attained the vertical peak force at 46% of the stance phase (Fig. 10). Generally, the 3rd toe bore the majority of load, with about 84% and 77% of the entire load during sand and ground trials, respectively, while the 4th toe carried only about 14% and 23% of the entire load during sand and ground trials, respectively. Although the curves of ground reaction forces of total foot, 4th and 3rd toes were all similar between substrates, the curve of the claw was obviously different, as the standard deviations under the claw were larger on sand surface than on solid surface. At toe-off, the vertical peak force shared by the claw was 1.5% and 2.4% of the entire load on loose sand and solid ground, respectively.

Ground reaction force in walking trials

The profiles of total vertical ground reaction forces in walking trials were slightly different from those recorded in running trails. The profiles in walking trials could be roughly divided into three parts: (1) a quick rise at 20% of the stance phase, (2) a slight increase at 50% of the stance phase, and (3) a rapid drop after about 70% of the stance phase (Fig. 11). The maximum total ground reaction forces on loose sand and solid ground were 1.3 and 1.1 times of the body weight, respectively, which occurred at 73% and 63% of the stance phase, respectively (Fig. 11A). In walking trials, the 4th toe, the 3rd toe and the claw successively attained the peak vertical ground reaction force. During the sand and ground walking trials, the vertical peak forces under the 4th toe were 10.7% and 22.8% of the vertical total forces respectively, and the contact durations between the 4th toe and the substrate were at 76% and 94% of the stance phase, respectively (Fig. 11C).

Results of statistical analysis

To analyze whether differences existed between two samples, we selected the peak pressures of different functional parts as indices. The differences in different media and different gaits between two samples were tested via one-way ANOVA. As shown in Table 3, no significant difference was found between the two samples (all p > 0.05), so the data of two samples were combined to analyze the significance of the gaits and the media.

Figure 11 Vertical ground reaction forces versus stance phase in loose sand and solid ground walking trials.

Vertical ground reaction forces of (A) total foot, (B) 3rd toe, (C) 4th toe, and (D) the claw were all normalized against body weight.

Table 3 Peak plantar pressures of two samples in different media and gaits.

		Solid ground	Loose sand	
	Sample	Walking	Running	Walking	Running	
Total	1	1.13 ± 0.290	2.14 ± 0.540	1.49 ± 0.200	2.55 ± 0.560	
2	1.12 ± 0.270	1.87 ± 0.390	1.34 ± 0.160	3.01 ± 0.810	
3rd toe	1	1.05 ± 0.300	1.64 ± 0.400	1.48 ± 0.110	2.17 ± 0.510	
2	0.99 ± 0.270	1.39 ± 0.320	1.34 ± 0.160	2.34 ± 0.090	
4th toe	1	0.24 ± 0.030	0.48 ± 0.150	0.15 ± 0.025	0.4 ± 0.130	
2	0.28 ± 0.055	0.47 ± 0.095	0.15 ± 0.053	0.4 ± 0.060	
Claw	1	0.041 ± 0.006	0.047 ± 0.030	0.014 ± 0.003	0.02 ± 0.010	
2	0.032 ± 0.018	0.047 ± 0.020	0.016 ± 0.016	0.04 ± 0.020	
Notes.

No significant inter-individual effects were found at P < 0.05.

The values were all normalized against the body weight.

Two factors were analyzed: gait mode (walking vs. running) and media (solid ground vs. loose sand). The data were analyzed using repeated-measures two-way ANOVA. The main effects of gait mode and media were analyzed for each part (total, 3rd toe, 4th toe and claw), and illustrated in terms of F value and p value (Table 4).

Table 4 F value and p value with consideration of interaction.

	Media	Gait mode	Interaction	
	F value	P value	F value	P value	F value	P value	
Total	11.87	0.0018	52.7	6.65 ×10−8	2.49	0.12	
3rd toe	25.49	2.42 ×10−5	36.89	1.5 ×10−6	2.47	0.12	
4th toe	8.78	0.0061	58.71	2.39 ×10−8	0.42	0.51	
Claw	7.84	0.0091	4.57	0.041	0.27	0.6	

As showed in Table 4, media and gait mode are significant independent factors, but their interaction is not significant. Thus, it can be stated the main effects of both media and gait modes are significant, but the effect on the outcome of the change in gait mode does not depend on the media. To further detect their interaction, we drew interaction plots (Fig. 12).

Figure 12 Peak plantar pressures under different media and gaits.

(A) Total force, (B) 3rd toe, (C) 4th toe, (D) Claw. The values were all normalized against the body weight.

The population means of media and gaits were all significantly different (P < 0.05). However, the data in Fig. 12 suggest only a weak interaction between media and gait mode, so we recalculated the effects of the two factors without interaction.

The population means of media and gait mode were still significantly different without consideration of interaction (p < 0.05; Table 5).

Discussion

The dynamic pressure distribution and the COP path during solid ground running or walking trials are consistent with a previous study (Schaller et al., 2011). This consistency in the profiles of ground reaction forces suggests the 3rd toe supports most loads and the 4th helps to maintain body balance. However, there are some differences in the COP path and the magnitude of ground reaction forces between the two studies, which should result from the differences in the collected velocities. Furthermore, our study is unique that we identified three high- pressure regions under the digital joints of the 3rd toe, which suggest the relation between the ground reaction force and the plantar skeleton, as well as the importance of digital cushion in protecting the plantar skeleton and soft tissues.

Table 5 F value and p value without consideration of interaction.

	Media	Gait mode	
	F value	P value	F value	P value	
Total	11.29	0.0021	47.81	1.34 × 10−7	
3rd toe	24.25	3.12 × 10−5	32.75	3.41 × 10−6	
4th toe	8.96	0.0055	61.2	1.25 × 10−8	
Claw	8.04	0.0082	4.4	0.044	

Locomotion on loose sand

During loose-sand running, the touchdown of the 3rd toe resulted in a larger heel contact area and a smaller attack angle, compared with solid ground locomotion. The attack angle is between the cross-sectional plane of the 3rd toe and the sand surface. As reported, the vertical reaction force of intruding into granular materials (Fz) can be computed as follows (Li, Zhang & Goldman, 2013): Fz=δ×A×Z

where δ is the vertical stresses per unit depth, A is the projected intruder surface area, and Z is the intrusive depth. For a given granular material, δ depends on the orientation and moving direction of the intruder. Although the relationship between δ and the intruder behavior is complicated, it is clear a smaller attack angle would increase the reaction force (Li, Zhang & Goldman, 2013). Thus, the ostrich toe orientation with a smaller attack angle during touchdown on loose sand could generate a stronger reaction force to support its body load. The phalanxes of the 3rd toe left the ground in order, which resulted in a smaller Z. An additional force, which was derived from the arch of the 3rd toe, not only enlarged the contact surface but also solidified the sands under the 3rd toe.

During push-off, the 3rd toe did not slip on sand, but the claw continued to intrude into sand. This phenomenon differed from the foot kinematics of zebra-tailed lizard moving on sand, in which the feet paddled through granular materials, generating force with huge energy loss (Li, Hsieh & Goldman, 2012; Hsieh & Lauder, 2004). The no-slip movement of ostrich toes and claw could be explained by the mechanism of granular material solidification. This mechanism was revealed by the locomotion of sea turtles on granular surfaces, in which the sea turtles created a hard sand region behind its flippers that remained no-slip locomotion (Mazouchova et al., 2010). Hence, we estimated the reaction force for advancing locomotion was partly supplied by the solid properties of sands under the toes. Furthermore, our experiments indicate the maximum friction coefficient of a plate could reach 0.65 before slipping during moving on granular medium (Nasuno, Kudrolli & Gollub, 1997). Therefore, the combination of the no-slip toes and the dense papillae covered on plantar surface may lead to acceleration because of considerable friction.

Our investigation suggests the 4th toe could correct the possible imbalance of the center of body weight on solid ground. Compared to the 3rd toe, the 4th toe shares less loads during the locomotion (particularly walking) on loose sand, which might result from the compensation of balance provided by the 3rd toe with its elliptical morphology. When the 3rd toe is intruding into loose sand, the lateral part could utilize the reaction of sand particles by compressing the sands to reduce the load of the 4th toe (Marvi et al., 2014).

Locomotion on solid ground

Repeated transient loading of foot strike, which occurred when the foot touched the ground and then transmitted upward the skeleton and soft tissues, might cause fatigue damage and accelerate running injuries (Whittle, 1999; Daoud et al., 2012). However, to eliminate the disadvantage of foot strike, the ostrich utilized a special strategy in long-distance running, which had been revealed by human kinematics examination. Abundant evidences show the habitually barefoot endurance runners as well as the running endurance Homo erectus landed on ground with the lateral foreparts of their feet, so as to avoid the transient forces (Lieberman et al., 2010; Bramble & Lieberman, 2004). These discrete points of the initial COP, located on the lateral distal part of the 3rd toe, indicate the ostrich landed on the ground with the lateral foreparts of the 3rd toe. During the impact process, the metatarsophalangeal joint flexed and the phalangeal joints extended as the toe moved under the control of tendons and muscles (Rubenson et al., 2011).

By flexing and extending these joints, ostrich toes partly translated the translational kinetic energy into the rotational kinetic energy of joints, which reduced the effective mass involved in the initial foot-ground contact phase and further decreased the magnitude of the impact transient force (Lieberman et al., 2010; Chia & Schmitt, 2005).

It has been suggested a part of foot strike is absorbed by the viscoelastic material located in the interface between the digital phalanx and the ground, especially the adipose cushion (Weissengruber et al., 2006). The viscoelastic material in ostrich footpad includes the adipose cushion, which consists of numerous adipocytes, a few other connective tissues, and abundant parallel collagen fibers (El-Gendy, Derbalah & El-Magd, 2011). Shock absorption of the viscoelastic material in ostrich footpad was brought by the deformation of fat cells and collagen fibers against the compression. Further studies on the mechanical property of the adipose cushion pad may help to comprehensively understand how the ostrich footpads save energy and reduce the magnitude of transient force (Gefena, Megido-Ravid & Itzchak, 2001).

Perspectives

Slight errors existed between the plantar pressure and the measured value in the loose sand, which were mainly due to the rolling and sliding of sand particles, and the extrusion and collision between sand particles. However, there is no proper test technology for measurement of large animal plantar pressures, when the large animal running or walking on loose sands. We laid 4-cm-thick sand on the pressure plate and tested the plantar pressure could be tested when an ostrich walked or ran on loose sand. Despite some errors in the quantitative measurement of ostrich plantar pressure, this method could be used to analyze the changing rules of ostrich plantar pressure and its proportion in different plantar parts when the ostrich was travelling on sand. Compared with the change rules of ostrich plantar pressure on solid ground, the high efficiency mechanism of the ostrich foot travelling on sand could be studied. In the future, we will continue to study the effects of sand thickness on ostrich plantar pressure during sand travelling, and explore the method to accurately measure ostrich plantar pressure in sandy environment.

Conclusions

The COP trajectory originated from the center of the 3rd toe when the ostrich ran on loose sand, which was different from the J-shape COP trajectory found on solid ground. The COP trajectory combined with the pressure distribution of each functional part and the motion of toes, indicating ostrich toes did not play the same functions on different substrates. On loose sand, the enlarged 3rd toe carried more body loads and provided sufficient traction by using the sand-solidification characteristics. The special arch and the well-developed claw of the 3rd toe also played a crucial role during locomotion on loose sand. On solid ground, the lateral part of the 3rd toe touched the ground in the beginning of stance phase, which prevented the skeletal system and soft tissues from harmful collision. On both substrates, the cushion pads of ostrich toes not only absorbed collision but also provided additional area to decrease pressure by deforming, especially at high speed. The morphology of toe improved the effect of traction and helped to maintain the balance of COM on sand surface.

Supplemental Information

Supplemental Information 1 Raw data

Click here for additional data file.

Supplemental Information 2 Statistical analysis with interaction

Click here for additional data file.

Supplemental Information 3 Statistical analysis without interaction

Click here for additional data file.

Supplemental Information 4 Video of ostrich running on loose sand

Click here for additional data file.

Supplemental Information 5 Video of ostrich walking on solid ground

Click here for additional data file.

Supplemental Information 6 Video of ostrich running on solid ground

Click here for additional data file.

Supplemental Information 7 Video of ostrich walking on loose sand

Click here for additional data file.

Additional Information and Declarations

Competing Interests

Author Contributions

Animal Ethics

Field Study Permissions

Data Availability

The authors declare there are no competing interests.

Rui Zhang conceived and designed the experiments, performed the experiments, contributed reagents/materials/analysis tools, reviewed drafts of the paper.

Dianlei Han analyzed the data, contributed reagents/materials/analysis tools, wrote the paper, prepared figures and/or tables.

Songsong Ma performed the experiments, analyzed the data, contributed reagents/materials/analysis tools, wrote the paper, prepared figures and/or tables.

Gang Luo conceived and designed the experiments, performed the experiments, wrote the paper.

Qiaoli Ji analyzed the data, contributed reagents/materials/analysis tools.

Shuliang Xue and Mingming Yang performed the experiments.

Jianqiao Li contributed reagents/materials/analysis tools, reviewed drafts of the paper.

The following information was supplied relating to ethical approvals (i.e., approving body and any reference numbers):

The Tab of Animal Experimental Ethical Inspection approved this study.

The following information was supplied relating to field study approvals (i.e., approving body and any reference numbers):

Ethical approval was given by the Animal Experimental Ethical Inspection, Jilin University.

The following information was supplied regarding data availability:

The raw data has been supplied as a Supplementary File.

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
