# Peer review of "Plantar pressure distribution of ostrich during locomotion on loose sand and solid ground"

_PeerJ, doi:10.7717/peerj.3613_

## Round 0.1 · original submission · Major Revisions

All three reviewers provide extensive constructive criticisms, and I think their points are very fair. We would need to re-review this paper if re-submitted, but it is going to need major revisions and re-writing in more fluent English (e.g. by a journal-writing service or colleague) to make it clearer. In addition, it will be very important to have a point-by-point Response to *all* of the reviewers' comments, so that it can be checked how well their critiques were dealt with. It is possible that the paper would still be rejected at that stage if this process was not adhered to rigorously.

·

Basic reporting

Some methodological details are missing including numbers of trials of each condition, the order in which conditions were presented, and randomization.

Experimental design

The design is acceptable.

Validity of the findings

No statistical results are reported, making the results purely descriptive in nature. I think that's fine provided the "statistically sound" criterion for publication is somewhat relaxed. Otherwise statistical analyses may be needed.

Additional comments

Protocol: how many trials were performed on solid ground and how many were performed with the substrate? Was the order of these two conditions randomly presented to each subject? I can’t seem to find this information in the Materials and Methods.

Kinematics: It appears from the Materials and Methods that joint kinematics were measured, but there do not appear to be any kinematics results. Please either add the kinematics results or remove the kinematics description from the Methods.

Methods, general: since you have measured transmitted forces and pressures is it possible that surface contact forces and pressures do not differ, and that what you have observed is due solely to soil force transmission mechanics? Please comment, preferably with literature references for methodological justification.

Measurement: With respect to Schaller et al. (2011), it appears that a key new result is a COP path difference as indicated in Fig. 6. However, I am unconvinced by this result for two reasons: (1) This is a qualitative interpretation of what appears to be two single trials. What about all of the other trials? (2) The substrate reduces the threshold of detectable pressure, so it is possible that the loss of the “J” shape is caused by measurement resolution change, and not by altered biomechanics.

Statistics: Data analysis appears to be primarily descriptive, based on representative trials or on mean / SD values. If PeerJ is willing to accept purely descriptive analyses then I don’t see a problem with the results, but if the journal prefers statistical analysis then please consider conducting statistical tests. From the Editorial criteria (https://peerj.com/about/editorial-criteria/) (number 3) it would appear that statistical analyses might be necessary. You may want to consider using a two-way repeated measures ANOVA design and a software package like spm1d (www.spm1d.org).

·

Basic reporting

The text requires substantial editing to be more understandable in general, a few specific comments are in general comments.

Experimental design

Some small comments on methods are included in general comments.

Validity of the findings

No comments.

Additional comments

This study examines differences in pressure and ground reaction force in ostriches moving over sand and solid ground. Locomotion over deformable substrates is not well-understood, so further research is welcome. The authors find differences in center of pressure location, trajectory, touch-down pattern, and ground reaction force shapes between speeds and substrates.

Before this study is ready for publication a few areas of concern should be addressed. The most important areas include improving the description of the results which describe center of pressure differences as well as clearing up angle of attack confusion. These and smaller issues are discussed in greater detail in the following comments.

Abstract- Move interpretation of angle of attack differences into discussion.
45-61- At least a brief mention of SLIP models should be made here.
119-120: "The traces..." Too vague; doesn't contribute information and should be included in the methods rather than here.
Introduction: The authors could emphasize more how locomotion on granular substrates is less well studied that solid substrates.
A credit to Schaller et al. 2011 near the beginning of the methods would be appropriate given that this study's methods seem to have been based on theirs.
123: It is worth noting whether the subjects were adults or juveniles.
Figure 2: Views of feet are mislabelled: b and c should be switched.
Figure 4: A histogram would show these data better.
195: Provide more detail on the angle between toes measurement needed- how was the axis of the toes determined.
206-216: These patterns are a major part of the paper's findings; make sure their description is clear. Lead the relevant section with the time being discussed- first or second half of stance, then compare/contrast the shape and location of the CoP, making sure to treat them separately before summarizing the patterns. Take more space if needed, and make sure sentences are clear and not too complicated.
Figure 6: The authors should consider highlighting the first and second halves of stance, or coloring the dots differently to ensure readers follow the descriptions. Also need a description of what pressure result is being shown.
251: Why is 8A not compared to 7A here rather than 6 which does not show the change through time?
Why are the GRFs from walking on solid ground not more obviously 2-humped? Schaller et al. 2011 results are similar, is it a result of using a pressure pad rather than a force plate?
According to the results from Schaller et al. 2011, some of the trials recorded here as runs would include grounded runs. This is worth mentioning.
318: "running" -> "walking"
336-337: The relationship between GRF and the skeleton alluded to here should be made explicit.
341: Attack angle/angle of attack appears here but not in results. Are the results inferred from the timing of toe pressure readings or are they from observed kinematics? Additionally, the definition of attack angle given here seems likely to be wrong; an angle created by the sagittal plane of the 3rd toe would be heading/compass direction, I believe this should be a medio-lateral plane. Further, the authors state that the attack angle in sand is smaller than on solid substrate, but on solid substrate the toes and claw hit closer to the same time which would suggest a smaller angle of attack, at least both by an intuitive sense for what angle of attack should be as well as how it is defined and used in Li et al. 2013. The authors may be using attack angle when they mean angle of intrusion (see Li et al. 2013). Make sure terminology is consistent and clearly defined.
348-350: This seems contradictory, how can the foot produce more force AND not sink as deep if the area is reduced? It seems like depth should increase as force increases and area decreases.
350-352: Seems to be saying the arch provides additional support in sand but the overall area of contact both of the foot and digit III is smaller in sand.
366-367: I agree for walking, but for running the 4th toe only goes from 23 % to 14% and has a similar peak. If the contribution in sand is going to be discounted for running, a stronger argument needs to be made why ~2/3 contribution is insufficient to perform the stabilizing role.
380: I think the toes extend while the TMP flexes: "The interphalangeal joint excursion mirrors that of the TMP joint during stance"- Rubenson et al. 2011
Raw Data: I don't think that QQ¢Ï-+20141227212259.png [filename is probably mistranslating into roman characters] is meant to be in the 0802A1 folder.

A supplemental movie of representative sand and solid trials would be a good addition to this manuscript, the same trials shown in figure 7 and 8 would make sense.

·

Basic reporting

This manuscript reports on walking and running of ostriches on two substrates: a hard substrate and a granular one (sand). The data collected consist of plantar pressures and high-speed video using three cameras yielding 3D kinematics.
The main focus of the manuscript is on a comparison of the plantar pressures in the four conditions.
Overall the manuscript certainly has the potential to contribute to our knowledge but several issues outlined here would need to be resolved.

Even though the kinematics part has been described in some detail in the text and in figures, unfortunately the manuscript does not report on them except in the Methods section lines 180-185. In the results no details are provided even for speeds, which I think they are crucial in order to provide context to the roll-off patterns and “ground reaction forces”. I addition, speed measures combined with stance times (which are known from the pressure plate) would yield duty factor which will differ between walking and running, but potentially also between substrates.
The authors might want to publish full 3D kinematics in a separate paper. In that case I would reduce detail on the 3D analysis in the Methods section, but still include at least speeds and duty factors in the current paper.

In contrast to the kinematics, which should be available have not been reported, ground reaction forces (GRF’s) _have_ been reported even though they have not been directly measured. I am aware that numerical integration of pressure over area should, in theory, yield ground reaction forces and I think that is how these have been obtained, directly from the pressure plate software (lines 193; no further details provided). There are at least two caveats with this. Firstly, by definition only vertical GRF’s would be obtained, not shear forces (fore-aft and left-right). Vertical GRF’s will certainly be larger than shear forces, in magnitude, but esp. the fore-aft force is important in braking and propelling the body and the capacity to deliver this force may be affected by a loose substrate such as sand (this is briefly touched upon in line xxx). Secondly, accuracy of the pressure sensors is inferior to that of force plates and integrating pressures will at best lead to a fairly good estimate of vertical GRF with a well-calibrated plate (interestingly, pressure plates are often calibrated by putting them on top of a force plate and comparing the output). Whilst I do not object at all to the presentation of these “GRF’s” I think the manuscript should provide some more detail on the GRF calculation and acknowledge the caveats in order to remove the impression that the GRF’s presented are highly accurate, as they would have been when measured with a dedicated force plate.

The manuscript style is overall very understandable and clear but the English language needs checking by a native speaker.

Reporting is rather descriptive. I do not object to descriptive papers but the material presented here lends itself to more functional interpretations. This has been done to some extent in the Discussion, but this is very brief and might be a bit random with relatively high detail on some aspects (e.g. lines 343-352) while some obvious related work has not been discussed. I think, for example, about work by Falkingham and colleagues on the mechanics of footprint generation in granular media. There is a body of literature on theropod locomotion which I think is worth considering in order to providing more relevance to the results from the current study. I think the paper would benefit from the inclusion of a few clearly stated research questions or hypotheses in the Introduction, and a Conclusion with clearer take-home messages (i.e. more about what the findings mean than a description of observations such as the shape of the CoP trajectory).

Figures are relevant and of sufficient quality. Some legends need improving or clarification. For example:
- Fig 2: what is “Isometric view”?
- Fig. 4: what exactly is the vertical axis, “Quality percentage”? This should be made clear here or in the text. Alternatively the figure might be skipped and average +/- s.d. of particle size presented. This might give similar or even more useful information to the reader whilst saving some space. Details on how particle size was determined are lacking.

Some minor issues:
- Lines 45-48: walking (when defined as a duty factor >0.5) is not necessarily dynamically equivalent to inverted pendulum and running is not necessarily equivalent to spring-mass (see many papers from researchers like Biknevicius, Full, Hutchinson, Kram, Seyfarth, etc.)
- It should be made clearer throughout the manuscript about which species statements are made if they are not general. For example, lines 372-375, this is about humans but since the previous sections were about ostriches this might be somewhat misleading.
- Line 75, since solidification is an important concept in the paper I think it merits to be introduced in some detail (a few lines).
- Line 99: how does a long limb enable high stride frequency? All else being equal, a long limb implies high resistance to rotational acceleration (high moment of inertia).
- Lines 119-120 “The traces were analyzed qualitatively to add some necessary data.” This phrase is not very meaningful and I suggest either deleting it, or clarifying which data it is about and why the resulting data are necessary.
- Line 123: _every_ part of their feet? Please specify which measurements were taken or delete if irrelevant.
- Lines 162 etc.: please specify spatial resolution of the cameras.
- Line 188: please check if sensor size is correct, or add “approximately” if necessary.
- Line 197: what is an “ordinary” camera? ;-)
- Line 375: what is the disadvantage of a foot strike? Do you mean heel strike? Even so, the question remains.

Raw data are available but they do not come with documentation to explain what the files are, as far as I can see. Furthermore the files have Chinese characters as column headers so for most readers the raw data are not useable at this point. All files need to be in English with clear headers and a single “manual” document outlining what the files in the raw folder are.

Experimental design

The study was performed on a limited number of individuals (2 individuals for the experiments, one different individual for MRI scanning). Basic subject data need to be added; the only information currently provided is that the subjects are adult females. Body weight is certainly available as it was used to normalise GRF.
No mention is made about the number of trials collected per individual and per substrate type. This is an important omission which needs to be corrected. Biological data are always variable and it is necessary to understand how large this variation is in the current data set especially since only two subjects were studied and since many results consist of “typical examples” or averages and means where it is not clear which data went into the calculations and in all cases N is not given.
The setup itself is very appropriate and of more than adequate length (80 m) for ostriches.
Detail is needed about how the pressure plate was calibrated (Line 199) as this is not always trivial and might have an impact on the numerical results.

Validity of the findings

The strength of the conclusions and the overall quality of the paper will depend to a large extent on the amount of data presented.

I suggest reporting speed, Duty factors (and potentially but not necessarily kinematics).

The effect of sand depth (4cm) on the data merits some discussion. I feel that 4 cm might be a good compromise between deep enough to be realistic (the natural substrate being “infinitely” deep for all practical purposes) and shallow enough to allow recording of pressures (less would be better). Nevertheless, it is almost without any doubt that different depths will result in different results and this observation merits at least a paragraph in the Discussion. For example, the plantar surface area values presented in Table 1 for the sand are in my opinion not true plantar surface areas but rather how pressure is transmitted down into the substrate onto the pressure plate, where the forces will be spread over a larger area than where they are produced at the foot-sand interface.

The “ground reaction forces” presented in Fig. 10 are puzzling. If animals have no overall vertical displacement (which I’m sure is the case over the 80-m long runway) then their average vertical impulse over a stride must always equal body weight. Judging by the area under the curves, this value (presented in _absolute_ body weights) seems lower on the solid surface but the horizontal axes are _normalised_ for stance duration. I can only make sense of these plots if the solid ground involves a longer duty factor (longer _absolute_ stance duration), which is an extra reason why i think such data need to be presented.

More statistics information is required; it had not been dealt with in the Methods and it seems that tests were not performed at all, e.g. to test for significance between conditions (substrates, gaits) or between individuals.

The Discussion includes interesting ideas (e.g. Lines 348-352, another potential reference could be water running in basilisk lizards). Presenting some kinematics of toe motion might strengthen these ideas.

The Discussion would benefit from a critical evaluation of shortcomings or potential issues (some of which outlined in this review).

---

## Round 0.2 · Major Revisions

Two of the reviewers have responded with moderate to major corrections needed. The English clarity still needs improvement. The statistical analyses need a major re-think and justification-- on that issue alone, the paper might be rejected on resubmission if it is not convincing (it is not right now). However, the reviewers do see hope for this manuscript, as do I. It will not be reviewed again-- as editor I will decide based on the MS with Tracked Changes and the Response to Reviewers (which should match in what is said in the Response to be done + actually done in the MS), whether the paper can be accepted or must be rejected at that point (it came a little close to rejection here). So please be extremely thorough in your final revisions.

·

Basic reporting

No comment

Experimental design

The actual design is a two-way repeated-measures ANOVA but has been analyzed using one-way ANOVA (see comments below).

Validity of the findings

No comment

Additional comments

Thank you very much for your thorough responses.

Main comment

Experimental design. Two factors were analyzed: GAIT MODE (walking vs. running) and MEDIA (solid vs. loose ground). This is a two-way repeated-measures design, not a one-way (non-repeated measures) design. Please repeat the analyses using repeated-measures two-way ANOVA. For each variable (total, 3rd toe, 4th toe, etc.) please report (1) the main effect of GAIT MODE, (2) the main effect of MEDIA, and (3) the interaction effect between GAIT MODE and MEDIA. Instead of reporting "significant" and "not significant" please report both F values and p values for all effects. Since many variables were analyzed please also consider adopting a correction for multiple comparisons.

Specific comments

Instead of “2 samples” please consider using “two samples”.

Abstract, Line 24. Instead of “The influence of gait on the peak pressure of the claw” I suggest: “Gait mode did not significantly affect claw peak pressure (p = 0.xxx)”. If you agree, please consider changing “gait” to “gait mode” throughout the manuscript when discussing walking vs. running. Also, please report an actual p value or p value range. For example: (p > 0.145).

Table 1. Instead of “effective number of tests” and “Tests used for statistical analysis” please consider using “Number of trials collected” and “Number of trials analyzed”, respectively.

·

Basic reporting

Writing still requires improvement

Experimental design

no comment

Validity of the findings

no comment

Additional comments

The authors have generally responded to my comments acceptably. Before publication, I do have a few remaining issues, some of which stem from my initial comments not being sufficiently clear, so I have clarified my concerns here. I agree with reviewer #1 that literature on theropods in granular substrates should at least be mentioned, and that the conclusions are generally specific, and would be improved by broadening, such as general implications for locomotion on loose substrates.

Detailed comments:

204-225- Contrary to the rebuttal, it does not appear that this section has been edited or changed.
Figure 6- I realize my wording was too vague in my first review. The colors should have a legend or be explained in the caption.
353- Delete the ",since there was no letter b in the table" clause, "No significant inter-individual effects were found" is sufficient.

In my original comments I wrote "According to the results from Schaller et al. 2011, some of the trials recorded here as runs would include grounded runs. This is worth mentioning." You replied "Running trials on both substrates were limited to the similar velocity period of 2.5-4.1 m•s-1. In the same way, walking trials were identified when the velocity period was in 0.7- 1.7 m•s-1. The grounded running gait was not singly involved in the present research. Grounded running and running gaits are analyzed together in our manuscript."
To be clear, I have no problem with binning grounded runs and running with an aerial phase together. However, my point is that Schaller et al. 2011 do make a distinction and do not include grounded running in most of their results: "As our primary interest was to identify clear differences in phalangeal pressure distribution during walking and running, trials of intermediate-speed grounded running were also disregarded in pedobarographic analysis." (p. 1125). Therefore I think it is worth explicitly noting this difference for readers who wish to compare your results with those from Schaller et al. 2011.

---

## Round 0.3 · Minor Revisions

As stated before, the MS will not be re-reviewed. I see only minor fixes needed at this point to make the MS publishable. Another pass at polishing the English would help, but as it stands the paper is legible enough. Some more important changes please:

Line 136 "with unbroken feet" better to say "with healthy feet".
Line 174 please state that grounded running was not studied.
Figs 3,6 please ensure these are large enough to be legible.
Fig 5 "ostrich plantar" say "plantar surface of the foot" or similar.
Fig 10,11 captions clarify that these are vertical ground reaction forces.
Table 3, Fig 12, and others: please ensure that units (e.g. N/cm^2??? It is not clear what is shown but values are low...) are always labelled.
lines 445-446 do you mean 4th not 3rd toe here? Please check.
I urge moving the "Perspectives" section to be before the "Conclusions".

Please make these amendments and resubmit and I will accept the paper. Do consider getting the English checked one final time though, please. But the paper has improved a lot and I think the reviewers' comments have been sufficiently addressed.

A final point: in the principles of open science, it would be ideal to provide all of your raw and/or processed data (e.g. all numbers subjected to statistical analyses) as Supplementary Information or in an open data repository (e.g. Figshare). This fits the principles of PeerJ best in that it makes the science reproducible. Please do this.

---

## Round 0.4 · accepted · Accept

Thank you for the attentive revisions. I have checked the Rebuttal and the Tracked MS and am satisfied. It is good that you provided your data. Congratulations on having your paper accepted!